# Revealing Natural Intracellular Peptides in Gills of Seahorse *Hippocampus reidi*

**DOI:** 10.3390/biom13030433

**Published:** 2023-02-24

**Authors:** Claudia Neves Correa, Louise Oliveira Fiametti, Gabriel Marques de Barros, Leandro Mantovani de Castro

**Affiliations:** 1Department of Biological and Environmental Sciences, Bioscience Institute, Sao Paulo State University (UNESP), Sao Vicente 11330-900, SP, Brazil; 2Biodiversity of Coastal Environments Postgraduate Program, Department of Biological and Environmental Sciences, Bioscience Institute, Sao Paulo State University (UNESP), Sao Vicente 11330-900, SP, Brazil

**Keywords:** peptidomics, seahorse, mass spectrometry

## Abstract

The seahorse is a marine teleost fish member of the Syngnathidae family that displays a complex variety of morphological and reproductive behavior innovations and has been recognized for its medicinal importance. In the Brazilian ichthyofauna, the seahorse *Hippocampus reidi* is among the three fish species most used by the population in traditional medicine. In this study, a protocol was performed based on fast heat inactivation of proteases plus liquid chromatography coupled to mass spectrometry to identify native peptides in gills of seahorse *H. reidi*. The MS/MS spectra obtained from gills allowed the identification of 1080 peptides, of which 1013 peptides were present in all samples and 67 peptide sequences were identified in an additional LC-MS/MS run from an alkylated and reduced pool of samples. The majority of peptides were fragments of the internal region of the amino acid sequence of the precursor proteins (67%), and N- and C-terminal represented 18% and 15%, respectively. Many peptide sequences presented ribosomal proteins, histones and hemoglobin as precursor proteins. In addition, peptide fragments from moronecidin-like protein, described with antimicrobial activity, were found in all gill samples of *H. reidi*. The identified sequences may reveal new bioactive peptides.

## 1. Introduction

Fish are the major component of aquatic fauna, living in a microbe-rich environment [1]. The seahorse is a marine teleost fish member of the Syngnathidae family that displays a complex variety of morphological and reproductive behavior innovations. It has been used in Traditional Chinese Medicine. It is believed that seahorse extract plays a role in diseases such as asthma, atherosclerosis, kidney diseases, infertility, erectile dysfunction, incontinence and skin diseases such as acne and persistent nodules [2,3].

Several studies have used dried seahorse powder hydrolyzed by proteases to obtain peptide extracts. Pharmacological investigation has revealed biological applications of these extracts. Antioxidant activity of enzymatic hydrolysates from the seahorse *Hippocampus abdominalis* has been reported in vitro and in vivo [4,5]. Although some protocols based on protein digestion are successful for the identification of bioactive peptides, in these preparations, the molecules are generated artificially in vitro.

A peptidome can be defined as the analysis of the peptides naturally generated in cells and tissues. It usually includes the following steps: sample collection, peptide extraction, fractionation, LC-MS/MS analysis, peptide identification and data mining [6]. Recently, with improvement in mass spectrometry and related proteomics analyses, it has become possible to sequence large numbers of peptides from complex mixtures. Peptides secreted from cells play roles in important physiological processes as signaling molecules (neuropeptides, hormones and growth factors), defense mechanisms (antimicrobial peptides, venoms and toxins) and modulating protein–protein interaction within cells [7]. Furthermore, fast protease inactivation techniques by heating, mainly microwave radiation, followed by molecular weight exclusion filters associated with LC-MS/MS, have revealed a constant set of peptides from cytosolic, mitochondrial and nuclear protein fragments in tissues of mice [7,8,9], zebrafish [10], human cell lines [11] and fungi [12], which have been termed intracellular peptides. Most of these peptides are intermediate products of protein metabolism and appear to be generated by the proteasome complex [13,14] that owns caspase-like (β1), trypsin-like (β2) and chymotrypsin-like (β5) proteolytic activities [6,15]. In addition, these intracellular peptides can play signaling roles after release from their parent proteins. Most bioactive peptides may be hidden in the sequences of functional proteins, such as hemoglobin and histone [16]. For example, Pep-H, a peptide derived from histone found in human brain tissue altered in schizophrenia patients, has shown protection from cell death [17].

The gills are organs that are constantly exposed to the surrounding water and are responsible for breathing, osmoregulation and excretion of nitrogenous waste products [18,19]. Furthermore, the gills are an immunocompetent organ due to the constant exposure to environmental challenges such as pathogens, pollutants and toxins. Several studies have demonstrated the presence of cells and molecules related to the innate immunity in gills, such as antimicrobial peptides (AMPs) and immunoglobulins, which play an important role in inhibiting pathogen invasion [20].

Antimicrobial peptides (AMPs) are related to the innate immune system of various organisms including humans, fish and plants [6,21]. AMPs usually consist of less than 100 amino acid residues, have a positive net charge and contain a hydrophobic region [22]. AMPs can kill bacteria, viruses and fungi by either disrupting their membrane integrity or inhibiting a cellular function. The contact of AMPs is initially due to an electrostatic and hydrophobic interaction with plasma membranes, usually involving pore-forming and non-pore-forming models [22]. Fish AMPs include cathelicidins, β-defensins, hepcidins, piscidins and histone-derived peptides [23].

Here, we performed a protocol based on fast heat inactivation of proteases to identify native peptides in gills of the seahorse *Hippocampus reidi*. A study of the Brazilian ichthyofauna showed that the seahorse of this species is one of the three fish species most commonly used by the population for medicinal purposes [24]. Peptide fragmentation data obtained by mass spectrometry were analyzed using a genome database of the recently characterized species *Hippocampus comes* [25], which allowed the identification of peptide sequences. In addition, the protein classes with the highest number of fragments found were discussed in relation to the antimicrobial potential of the peptide sequences. 

## 2. Materials and Methods

### 2.1. Animals

Three *Hippocampus reidi* adult specimens were obtained by hand while SCUBA diving in Ubatuba, São Paulo, Brazil (23°27′01.41″ S–45°02′09.00″ O), and kept in the Laboratory of Marine Proteins and Peptides of the Bioscience Institute of Sao Paulo State University. Artificial seawater was prepared using commercial sea salt and mixed according to the manufacturer’s instructions. Animals were housed in aquaria using artificial seawater (water temperature 25 °C) and fed twice a day.

This study followed the guidelines of the National Council for Animal Experimentation Control (CONCEA), permission of SISBIO (Protocol Number 78669-1), and was approved by the Ethics Commission for Animal Use (CEUA) at the Bioscience Institute of Sao Paulo State University (Sao Vicente, Brazil; Protocol Number 02/2021-CEUA).

### 2.2. Peptide Extraction

Seahorse peptide extracts were prepared as previously described [10]. Adult male seahorses were anesthetized with a lethal dose of MS 222 (100 mg/L) and taken to the microwave for 10 s to inactivate peptidase and proteases. Next, gills were collected in centrifuge tubes. Ten one-second sonication pulses (4 Hz) were applied to homogenize each sample in 1 milliliter of deionized water, which were then maintained at 80 °C for 20 min. After cooling in ice, HCl was added to a final concentration of 10 mM. The homogenates were centrifuged at 5000× *g* for 40 min at 4 °C. A Millipore centrifugal filter unit with a molecular weight cut-off of 10.000 Da was used to filter the supernatants. C-18-like Oasis columns (Waters) were then used to purify and concentrate peptides contained in the samples, which were then dried in a vacuum centrifuge. The extract obtained was stored at −80 °C for a subsequent peptidomic analysis.

In addition to identifying peptides with cysteine residues, 100 µg from a pool of gill samples was submitted to a reduction and alkylation reaction in order to identify cysteine residues in samples. For the reduction reaction, 5 mM of final concentration of DTT (Dithiothreitol) was added to the sample and incubated for 25 min at 56 °C. The alkylation reaction was performed with the addition of IAA (Iodoacetamide) at a final concentration of 14 mM and incubated for 30 min at room temperature and in the dark. After incubation, a quench of free IAA was performed by adding at 5 mM final concentration, being incubated for another 15 min at room temperature and in the dark.

### 2.3. Fluorescamine

Fluorescamine was used to determine peptide concentration at pH 6.8. The reaction was performed at pH 6.8 to ensure that only the amino groups of the peptides react with the fluorescamine, avoiding that free amino acids react with it. In short, 2.5 µL of the sample was mixed with 25 µL of 0.2 M phosphate buffer (pH 6.8) and 12.5 µL of a 0.3 mg/mL acetone fluorescamine solution, then vortexed for 1 min. Next, 110 µL of water was added, and a SpectraMax M2e plate reader (Molecular Devices, Sunnyvale, CA, USA) was used to measure the fluorescence at an excitation wavelength of 370 nm and an emission wavelength of 480 nm. The standard reference for the determination of the peptide concentration was the peptide 5A (LTLRTKL), which has a known composition and concentration.

### 2.4. Liquid Chromatography and Mass Spectrometry

The peptide mixture was suspended in 0.1% formic acid and analyzed as follows. An UltiMate 3000 Basic Automated System (Thermo Fisher®) was set up and connected online with a Fusion Lumos Orbitrap mass spectrometer (Thermo Fisher®) at the mass spectrometry facility RPT02H/Carlos Chagas Institute-Fiocruz, Paraná. The peptide mixture was chromatographically separated on a column (15 cm in length with an internal diameter of 75 μm) packed in-house with ReproSil-Pur C18-AQ 3 μm resin (Dr. Maisch GmbH HPLC) with a flow rate of 250 nL/min of 5% to 38% ACN in 0.1% formic acid on a 120 min gradient. The Fusion Lumos Orbitrap was placed in data-dependent acquisition (DDA) mode to automatically turn between full-scan MS and MS/MS acquisition with 60 s dynamic exclusion. Survey scans (300–1500 m/z) were acquired in the Orbitrap system with a resolution of 120,000 at m/z 200. The most intense ions captured in a 2 s cycle time were chosen, excluding those which were unassigned or had a 1+ charge state. The selected ions were then isolated in sequence and fragmented using HCD (higher-energy collisional dissociation) with normalized collision energy of 30%. The fragment ions were analyzed with a resolution of 50,000 at 200 m/z. The general mass spectrometric conditions were as follows: 2.3 kV spray voltage, no sheath or auxiliary gas flow, heated capillary temperature of 175 °C, predictive automatic gain control (AGC) enabled, and an S-lens RF level of 30%. Mass spectrometer scan functions and nLC solvent gradients were regulated using the Xcalibur 4.1 data system (Thermo Fisher®).

### 2.5. Peptide Identification

The raw data files were submitted to search against the NCBI database filtered for taxonomy *Hippocampus comes* using the software PEAKS Studio (version 8.5; Bioinformatics Solution, Waterloo, ON, Canada). The decoy-fusion method was used to search a decoy database in order to calculate false discovery rate (FDR). The following search parameters were considered: (a) precursor mass tolerance to ±30 ppm; (b) fragment ion mass (tolerance of ±0.5 Da); (c) variable modifications: oxidized methionine (+15.99 Da) and acetylation (+42.01 Da); (d) no enzyme specificity. The identified peptides were then sorted by their average of local confidence to select the best spectra for annotation, and they were filtered by FDR ≤ 0.1%.

## 3. Results

The MS/MS spectra obtained of *H. reidi* gills extracts allowed the identification of 1080 peptides from 396 proteins, of which 1013 peptides were present in all samples initially analyzed (Appendix A). Sixty-seven peptide sequences were identified in an additional LC-MS/MS run from an alkylated and reduced pool of gill samples. This run was performed to improve the detection of peptides containing cysteine residues (Appendix A).

Regarding the general characteristics of these peptides, the fragments ranged from 5 to 38 amino acid residues, with 80% having a length between 8 and 18 amino acid residues (Figure 1A). The precursor proteins of these intracellular peptides are mainly located in the cytosol (58%), nucleus (22%) and mitochondria (5%) (Figure 1B). In addition, most of the peptides found in the gills originated from ribosomal proteins (32%), histones (11%), proteins related to cytoskeleton (9%) and hemoglobin fragments (5%) (Figure 1C).

In order to investigate aspects related to proteolytic processing, the amino acid sequence of the precursor proteins and the peptides generated were analyzed. Most of them were fragments of the internal region of the amino acid sequence of the precursor proteins (67%). N- and C-terminal regions represented 18% and 15%, respectively (Figure 2A). About 70% of the N-terminal fragments were found have an N-terminal acetyl group. The most frequent amino acid in the N-terminal region of the identified peptides was alanine, followed by serine, leucine and lysine (Figure 2B). Rare cleavage sites include tryptophan and glutamine. In addition, arginine is the most common residue in the P1 position (60.19%), followed by lysine (21.12%) (Figure 2C).

The quantification data for each peptide sequence found in all gill samples was performed and is shown in Appendix A. The ratio of each peptide detected in one sample was calculated in relation to the average of the two other samples and the average ratios shown in Figure 3A. Of the 1013 quantified peptides, 72.8% had an average ratio between 1.0 and 2.0, 16.1% between 2.0 and 4.0 and the remaining (11.1%) distributed in different ratio ranges above 6 (Figure 3B).

## 4. Discussion

The main result shown here was the characterization of the peptide profile naturally present in the gills of the seahorse *H. reidi*. Most studies involving peptide extracts from seahorses have used enzymes for the preparation of hydrolysates, generating sequences artificially. Specific sample preparation methods and LC-MS/MS analysis can contribute to the predominant identification of peptides with distinct physicochemical properties [6]. Peptides in seahorse gills were extracted after microwave irradiation for rapid inactivation of proteases by heat, followed by extraction in cold acidic deionized water to avoid postmortem artifacts [7]. The features of peptides identified in this study, such as the fragment size, their subcellular distribution and protein classes, showed similarities with other peptidome studies that applied the same extraction protocol [10,12].

Most peptides within cells are generated during protein degradation by the large complex proteolytic proteasome that generates fragments within 5 to 22 amino acids [6,15]. This range of the sequences represented more than 90% of peptides found in gills of *H. reidi*. The three major catalytic activities associated with proteasomes are caspase-like, trypsin-like and chymotrypsin-like activities that cleave after acid, basic and hydrophobic residues, respectively [15]. Despite the similarities with other peptidome studies, with respect to the general characteristics mentioned previously, differences related to proteolytic processing were observed. Our findings show strong trypsin-like activity in peptides, with a predominance of basic amino acids in the P1 position of the cleavage site, in addition to the fragments being mostly from the internal region of the precursor proteins. These differences could be related to the functional aspects of the gills, as an organ that has contact with the external environment. A meta-analysis of human fluid peptidome demonstrated that, at least for serum and tears, there is a preference for Arg and Lys, positively charged amino acids, in the P1 position [26]. The presence of basic amino acid residues in the P1 position is common in peptides generated during the classical secretory pathway, such as neuropeptides and hormones [27]. The proteasome is well known for its role in generating antigenic fragments during adaptive immunity [28,29]. However, it has been suggested that the proteasome also produces antimicrobial peptides as part of the innate immune response, as shown with the intermediate filament keratin 6a (K6a), which is constitutively processed into antimicrobial fragments in corneal epithelial cells [30]. Thus, trypsin-like protease activity seems to have a significant contribution to the shape of peptides in gills of seahorse.

Quantitatively, most peptides (72,8%) showed an average ratio between samples of 1 to 2%. In a zebrafish brain peptidome study without treatments, variations in the average ratio of peptides were also observed, with some intracellular peptides showing a greater variation than the neuropeptides [10]. Furthermore, in our analyses, peptide fragments from the same protein that presented different ratios were observed, which may indicate a distinct proteolytic processing.

Mucosal barriers, such as the skin, gills and gut epithelia, provide a first line of defense against infection [31]. Antimicrobial peptides (AMPs) have been identified in gills of several species of fish [1,32,33,34]. Moronecidin is an AMP member of the piscidin family that has high salt tolerance and a broad spectrum of activity against microorganisms. The membrane disruption of moronecidin in microbes is due to the formation of a pore [31]. The alignment of moronecidin-like peptide, originally identified in *H. comes*, with other piscidins shows that the signal peptide is conserved, whereas the sequences of the mature peptide are variable [35]. A total of fifteen fragments from the moronecidin-like peptide were found in gill samples of *H. reidi* (Figure 4A). Among these sequences identified here, the fragments FFRNLWKGAK, KGAKAAFR, GAKAAFR and NLWKGAKAAFR are part of a peptide selected by in silico analysis of the *H. comes* genome and recently characterized as an antimicrobial peptide [35]. The sequence FFRNLWKGAK (Figure 4B) preserves amphipathic characteristics, such as α-helix formation, with hydrophobic residues on the same surface and also positively charged amino acid residues (Figure 4C), as described for piscidins [23,36].

Ribosomal proteins, histones, and hemoglobin were the intracellular proteins that had most sequences found in the gill peptidome. Antimicrobial activity was demonstrated for peptides from these intracellular proteins in many different species [16].

Peptide sequences from ribosomal proteins represented 32% of the fragments in the gills of the seahorse *H. reidi*, with many fragments of the different ribosomal proteins that constitute the 40S small subunit and the 60S large subunit of eukaryotic ribosomes. Of the 78 described ribosomal proteins that constitute these two subunits, 61 were detected in our data, of which 36 proteins belonged to the 60S subunit and 25 proteins to the 40S subunit. In Figure 5, the number of fragments found for each ribosomal protein is shown. Ribosomal protein L13 was the one with the highest number of fragments. Hurtado-Rios and collaborators [37] have recently discussed the role of ribosomal proteins as moonlighting proteins, which are those capable of performing more than one biochemical or biophysical function within the same polypeptide chain, highlighting them as natural antimicrobials [37,38,39]. Recently, a C-terminal fragment of the 60S ribosomal protein L27 was isolated from the skin of *S. asotus* and identified as AMP [40]. In another study, antimicrobial activity against *Bacillus megaterium*, *Escherichia coli* and *Candida albicans* was detected in an extract from the epidermal mucus of Atlantic cod (*Gadus morhua*). Ribosomal proteins L40, L36A and L35 were identified in fractions prepared by weak cation exchange chromatography together with reversed-phase chromatography and mass spectrometry [41]. In addition, in oyster gills from *Cassostrea gigas*, a fragment of 60S ribosomal protein L29, with 54 amino acid residues, was identified as AMP. Table 1 lists some identified sequences of ribosomal proteins with similar properties of antimicrobial peptides.

Histones are evolutionary conserved basic proteins, present in all eukaryotic cells. They are known to function in chromatin structure formation, nuclear targeting and regulation of gene expression [42]. Histone-derived AMPs have been identified in a number of fish species, with broad-spectrum activity against both human and fish pathogens, suggesting that complete sequences, N-terminal or C-terminal fragments are part of an ancient innate immune mechanism [43,44].

Histones H2A and H2B have been found in microsomes from gill epithelium of five species of primitive to advanced teleost fish, indicating that these proteins might be secreted to the extracellular environment [45]. In addition, histone H2A has been shown to be synthesized in excess, in amounts required for DNA packaging, and accumulates in cytoplasmic granules in gastric gland cells. Upon secretion into the gastric lumen, it is processed by pepsin C isozymes to yield buforin I [46,47], and proteasomal degradation of histone proteins increases after oxidative damage [48]. The peptide buforin II (TRSSRAGLQFPVGRVHRLLRK) derived from buforin I demonstrates a potent antimicrobial activity, killing the bacteria without cell lysis and having affinity with nucleic acids, apparently inhibiting the cellular function by binding to DNA and/or RNA [49]. A similar sequence TRSSRAGLQFPVGRVLR, identified as histone H2A-like found here, presents a hydrophobic ratio of 35%, total net charge of +4, four hydrophobic residues on the same surface, and has 80,75% of similarity with buforin II. Here, most of the peptide sequences found for H2A and H2B represented fragments of the N- and C-terminal of these proteins (Figure 6).

Peptide sequences derived from hemoglobin were 5% of fragments present in seahorse gills. Hemoglobin is an essential protein for maintaining cellular homeostasis due to its ability to bind and transport oxygen to the tissues. This protein has also been associated with immune response modulation, signal transduction and antimicrobial activity [50]. Hemoglobin has been known as a source of endogenous bioactive peptides that present several different functions [51,52,53]. Regarding innate immunity, the presence of antimicrobial peptide fragments of hemoglobin isolated from tissues such as skin, branchial epithelium and liver has been shown in fish. For example, an antiparasitic effect of hemoglobin-derived AMPs has been identified from the epithelium of the catfish *Ictalurus punctatus*, with changes in both the HbβP-1 sequence transcribed and translated in skin and gill epithelium against infection of *Ichthyophthirius multifiliis*, where the hemoglobin concentration expressed in vivo appeared to be similar to the antiparasitic concentrations measured in vitro [54]. Furthermore, a study in sea bass detected changes in Hb-LP gene expression by real-time RT-PCR in the gills and skin in acute crowding stress, but not in other tissues [55]. In addition, among the various fragments of the hemoglobin alpha chain found in *H. reidi* gill samples (Figure 7A), fifteen sequences were from a similar region corresponding to the peptide FAHWPDLGPGSPSVKKHGKVIM, derived from hemoglobin alpha in the liver of Japanese eel, *Anguilla japonica*, with strong antibacterial activities against Gram-positive or -negative bacteria [56]. One of these fragments showed a 56% similarity with this previously described peptide (Figure 7B).

Our study also verified the presence of peptides containing cysteines through an additional experiment with reduction and alkylation without trypsinization, because some classes of antimicrobial peptides, such as defensins, that are small, amphiphilic and cationic peptides, are usually rich in cysteines [57]. Of the sixty-seven peptides with cysteine residues found, ten sequences contained two residues and the others only one. However, vertebrate defensins are cationic antimicrobial peptides and have three pairs of disulfide bonds forming three intramolecular disulfide bonds [58]. Future investigations will be necessary to verify the antimicrobial activity of these sequences.

Taken together, it was possible to identify the presence of fragments similar to peptides with described antimicrobial activity, and here only this aspect was explored. However, other intracellular peptide fragments identified in this study may be related to other distinct biological functions, as it has been shown in recent years [52,59,60].

## 5. Conclusions

The gill peptidome of seahorse *H. reidi* obtained through fast heat inactivation of proteases revealed a set of peptides consisting of many fragments from intracellular proteins residing in compartments such as the cytosol, nucleus and mitochondria. Our data bring relevant information on the levels of these molecules, showing a particular behavior for each peptide, which presents variations from animal to animal, even before being submitted to an experimental condition. Most of the sequences found showed sizes and properties similar to peptidomes from tissues of other organisms where the same protocol was used. However, the proteolytic processing analysis showed differences, such as the prevalence of internal fragments of precursor proteins and the predominance of basic residues such as arginine and lysine in the P1 position of the cleavage site. These differences may be related to some functions of gills, such as innate immune response, which may generate possible peptide sequences with antimicrobial action. Furthermore, our results confirm data from the literature that demonstrate antimicrobial activity of peptide fragments of intracellular proteins. Recognition of patterns and identifying key peptides in seahorse gills could lead to a better understanding of processes such as immune response and proteolysis dynamics. The identified sequences could yield new therapeutic peptides as an alternative to the resistance presented by many pathogenic microorganisms to antibiotics available as treatment.

## Figures and Tables

**Figure 1 biomolecules-13-00433-f001:**
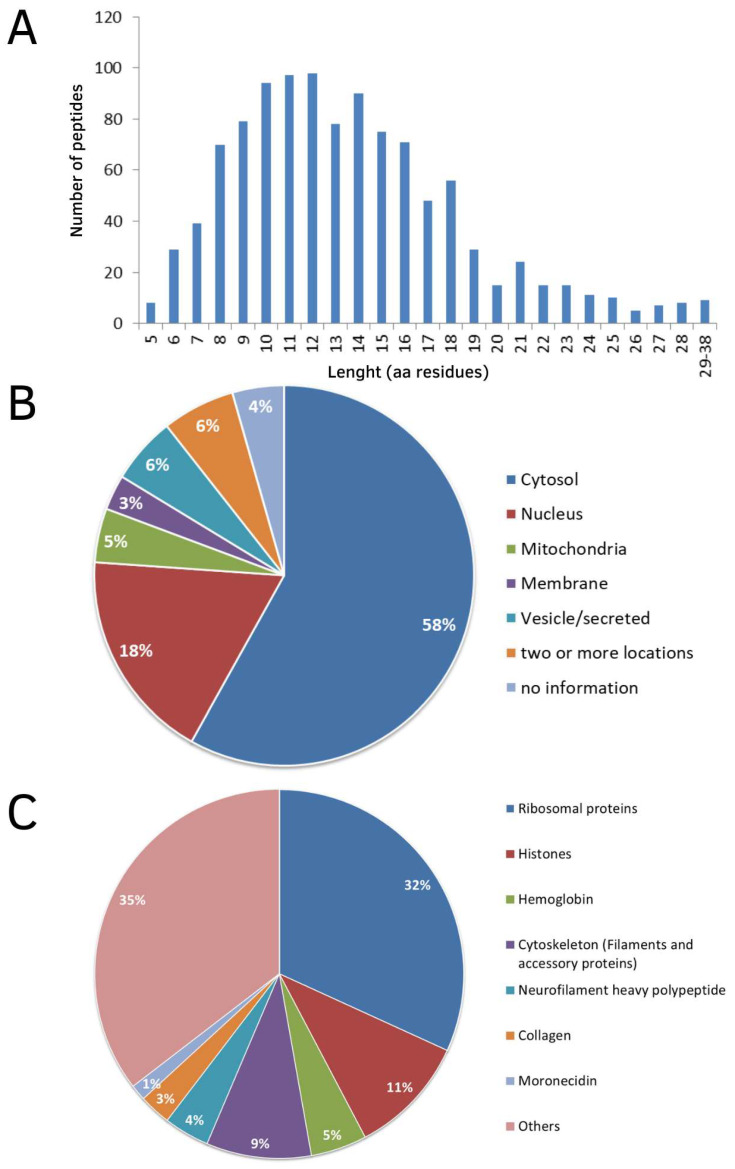
General features of peptide sequences found in gills of seahorse *H. reidi*. In (**A**), peptide distribution by length (amino acid residues). In (**B**), subcellular location of precursor proteins of peptides. In (**C**), classes of precursor proteins.

**Figure 2 biomolecules-13-00433-f002:**
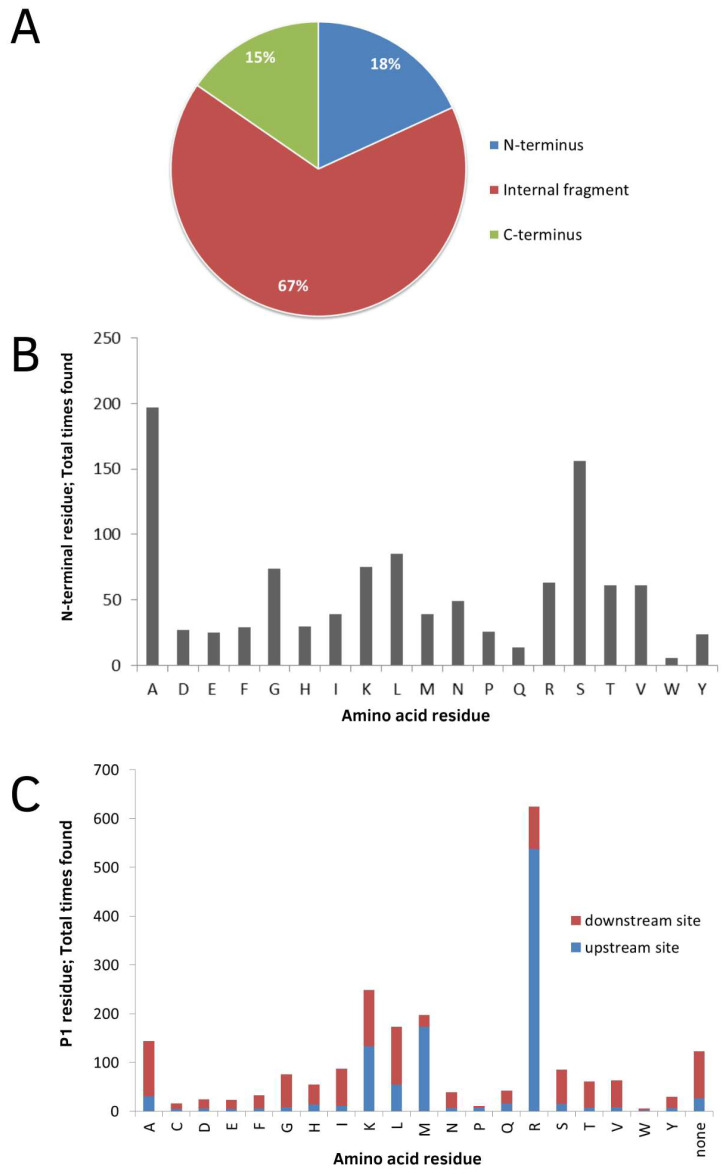
Proteolytic processing of peptides in gills of *H. reidi*. In (**A**), analysis of the peptide location within the protein precursor. In (**B**), analysis of the N-terminal residues of the peptides; number of times each amino acid is located at the N-terminus considering the 1080 identified peptides. In (**C**), analysis of the cleavage sites required for the formation of the observed intracellular peptides. The number of occurrences of each amino acid at the P1 position of the cleavage site is indicated for both upstream and downstream sites.

**Figure 3 biomolecules-13-00433-f003:**
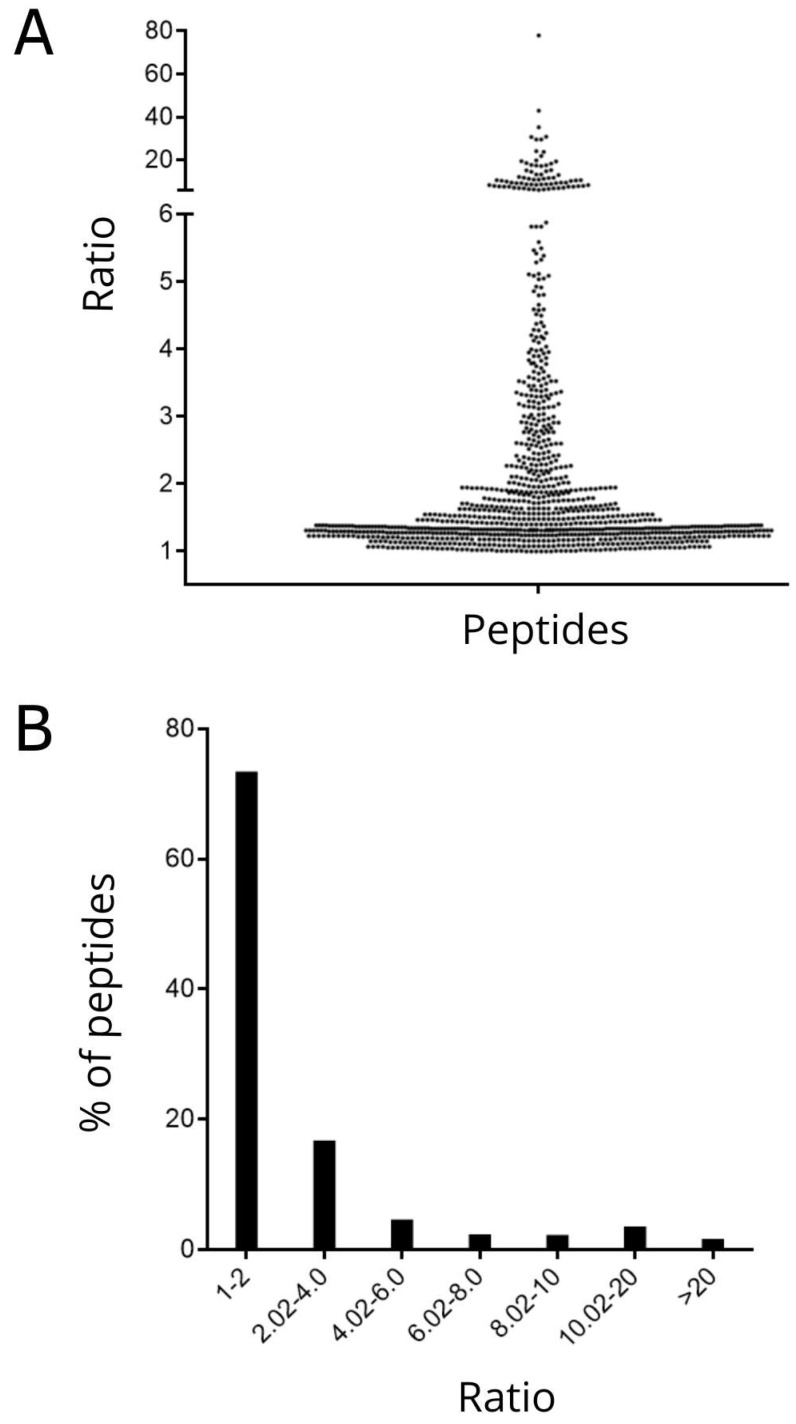
Average ratio of each peptide found in *H. reidi* gills (**A**). Percentage of peptides in different ranges of ratio (**B**).

**Figure 4 biomolecules-13-00433-f004:**
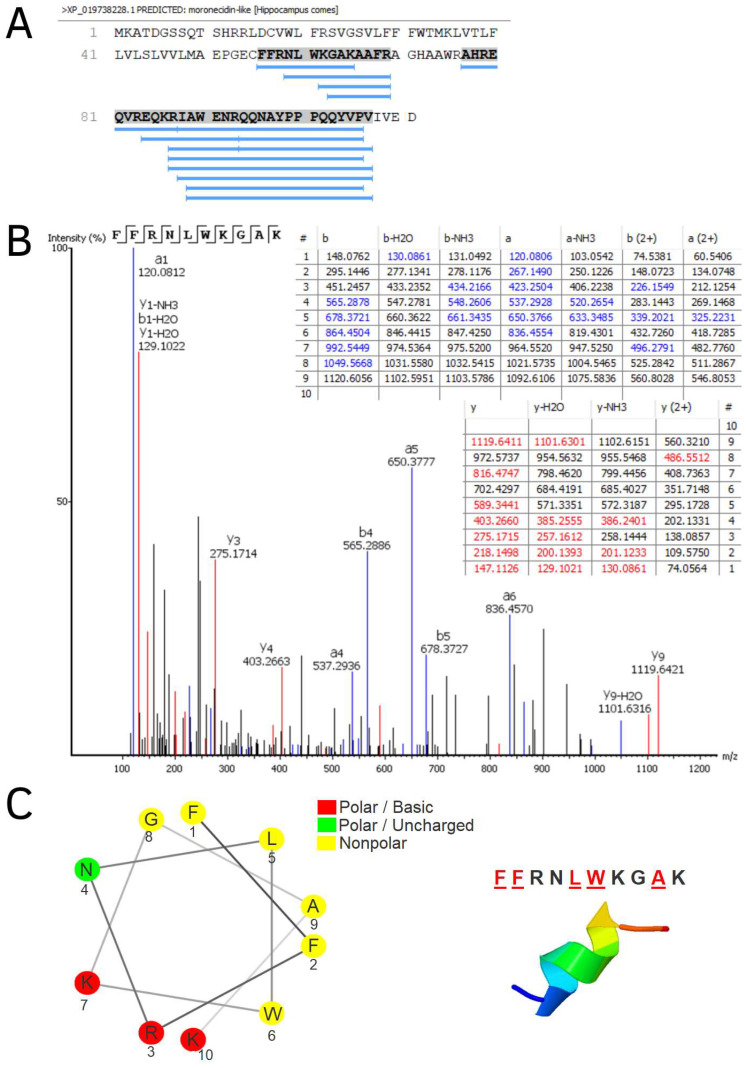
Moronecidin-like found in *H. reidi* gills. In (**A**), location of peptide fragments in the precursor protein sequence. In (**B**), MS/MS spectrum of the sequence peptide FFRNLWKGAK. In (**C**), presence of hydrophobic (yellow) and basic (red) residues, performed using the online tool NetWheels (https://netwheels.herokuapp.com/; accessed on 25 August 2022) and alpha helix prediction analysis built using I-Tasser (https://zhanggroup.org/I-TASSER/; accessed on 25 August 2022).

**Figure 5 biomolecules-13-00433-f005:**
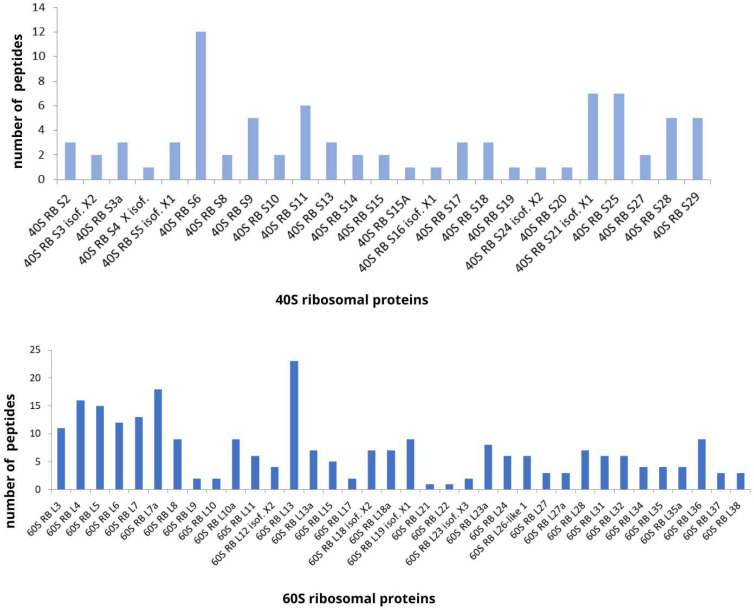
Number of peptides identified for each ribosomal protein that constitute the 40S small subunit and the 60S large subunit of eukaryotic ribosomes.

**Figure 6 biomolecules-13-00433-f006:**
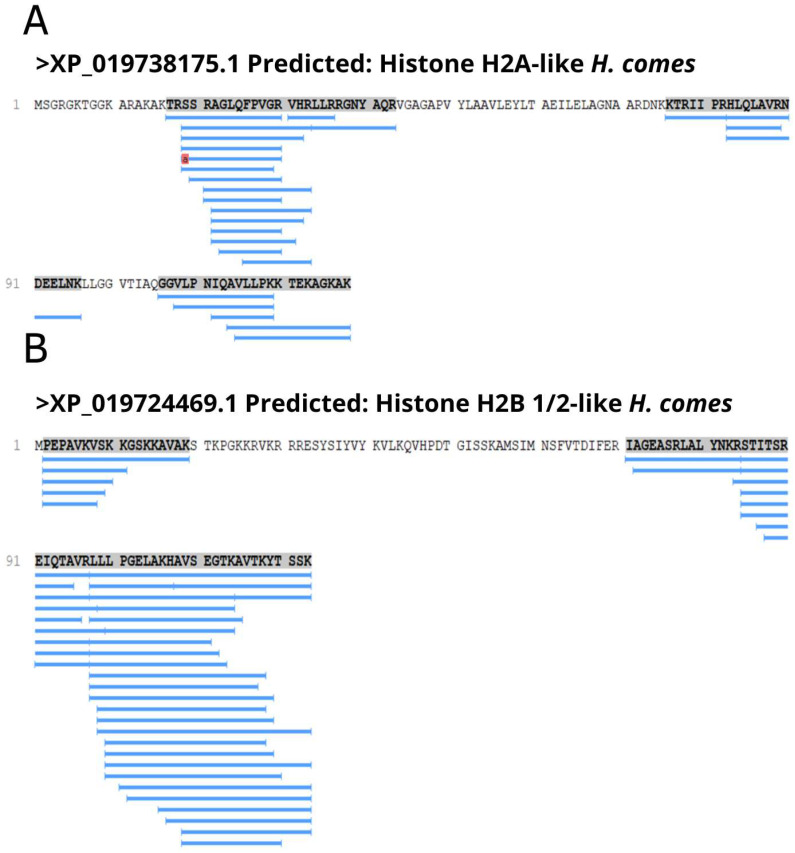
Histone-like derived peptides identified in gills of *H. reidi.* Location of peptide fragments (underlined in blue) in the amino acid sequence of the predicted histone H2A-like (**A**) and histone H2B-like protein (**B**).

**Figure 7 biomolecules-13-00433-f007:**
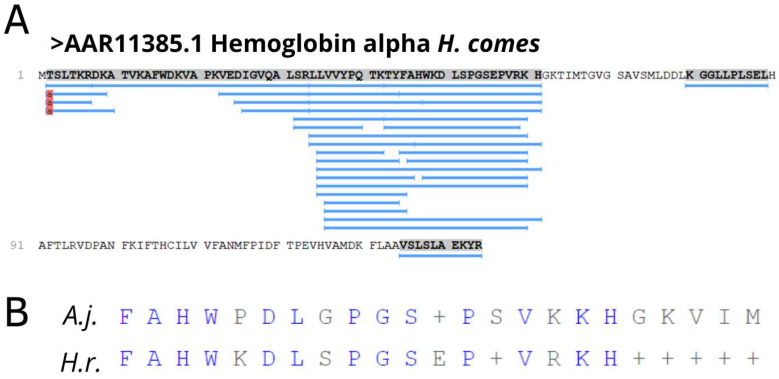
Hemoglobin alpha chain derived peptides identified in gills of *H. reidi*. In (**A**), location of peptide fragments (underlined in blue) in the amino acid sequence of the precursor protein. In (**B**), sequence alignment of a peptide sequence found in the gills of *Hippocampus reidi* (H.r) and a peptide sequence described with antibacterial action in the liver of *Anguilla japonica* (A.j.).

**Table 1 biomolecules-13-00433-t001:** Sequences of ribosomal proteins with similar characteristics to AMPs. Hydrophobic residue (HR) on the same surface.

Sequence	Protein	HR	Net Charge	Gravy	HR1
TVGVQPAADGKGVVVVIKKR	60S ribosomal protein L28	45%	3	0.33	0
SQGTRDLDRIAGQVAAANKKSA	40S ribosomal protein S19	36%	3	−0.72	5
AGRGFTLEELKAAGIHKKTAR	60S ribosomal protein L13	38%	3.25	−0.54	5
VRKLYDIDVSKVNTLIRPDGEKKAYVR	60S ribosomal protein L23a	33%	3	−0.64	6
NFGIGQDIQPKRDLTRFVKWPR	60S ribosomal protein L7a	32%	3	−0.99	3
PKGKKAKGKKVAPAPVVAK	60S ribosomal protein L7a	37%	7	−0.68	0
SANRAVVGVVAGGGRIDKPILK	60S ribosomal protein L8	45%	3	0.32	8

## Data Availability

The data is contained within the article and mass spectrometry supporting information (raw files) can be downloaded at: https://massive.ucsd.edu/ProteoSAFe/dataset.jsp?task=274624a43f5d4e9097dc1311d5a0d140.

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
