# Peer review of "Revealing Natural Intracellular Peptides in Gills of Seahorse Hippocampus reidi"

_biomolecules, 2023, doi:10.3390/biom13030433_

Round 1
Reviewer 1 Report
The manuscript entitled “Revealing natural intracellular peptides in gills of seahorse
Hippocampus reidi” by Correa, et al gives an overview on gill peptidome of seahorse seahorse Hippocampus reidi. the study is interesting however few points must be considered:
English must be checked by native English speaker
Introduction is too lengthy and should be shortened
References need update
The conclusions part is very poor, should be elaborated to show the impact of the study and future directions.
Author Response
Comments and Suggestions for Authors
The manuscript entitled “Revealing natural intracellular peptides in gills of seahorse Hippocampus reidi” by Correa, et al gives an overview on gill peptidome of seahorse seahorse Hippocampus reidi. the study is interesting however few points must be considered:
Thank you for your very nice and critical comments, that were all very appreciated. Below, please, find our attempt to answer your main questions.
- English must be checked by native English speaker
Answer: English revision was performed by native English speaker
2. Introduction is too lengthy and should be shortened
Answer: Thank you for the suggestion. Our intention was to clearly present the topics related to the study, however we understand that the introduction was long. We have summarized some basic information and also removed some parts of the text that we feel do not compromise this section of the manuscript.
3. References need update
Answer: Thank you very much. Most of the studies mentioned in the text were published in the last ten years, mainly the articles relating to the peptidomics field. However, we checked the literature again and included new references to some points in the discussion where older articles are mentioned. In this context, new information and their respective references were added to the manuscript.
4. The conclusions part is very poor, should be elaborated to show the impact of the study and future directions.
Answer: the conclusion has been improved.
Reviewer 2 Report
The authors show a bioprospecting study through peptidome analysis in Hippocampus reidi gills. The study is interesting, and the manuscript is well prepared, congratulations. I think it's worth being published, the only comment I have is that in table 1, the comment says hidrofobic, instead of hydrophobic. I hope that in the future they can verify the activity of the peptides found.
Author Response
Answer: Thank you very much. We are currently investigating the antimicrobial and antioxidant activity of some identified peptides in seahorse gills. We have corrected the word indicated by the reviewer.
Reviewer 3 Report
The manuscript by Correa et al described a set of peptides obtained through the fast heat inactivation of proteases in gills of seahorse Hippocampus reidi. The identity of peptides was confirmed by liquid chromatography coupled to mass spectrometry (LC/MS).
The manuscript lacks clarity and following issues need to be resolved before publishing:
1. Did author check the bioactivity of identified peptides?
2. There is no information provided on the quantification of obtained peptides.
3. At page 2 the author should revise the nomenclature of showing ions……Several proteins related with ion transporters are expressed in the membrane cells of the gills (NKA- Na+-K+ - ATPase; NKCCl- Na+-K+-2Cl- cotransporter; NHE-Na+-H+ exchanger; H-A-H+ATPase……
4. The quality of figures is poor.
5. In figure 3B, the author should consider labelling the peptide fragments (y3, y4 ...) on top of the sequence corresponding to the peaks observed in the mass spectra for more clarity.
Author Response
The manuscript by Correa et al described a set of peptides obtained through the fast heat inactivation of proteases in gills of seahorse Hippocampus reidi. The identity of peptides was confirmed by liquid chromatography coupled to mass spectrometry (LC/MS).
The manuscript lacks clarity and following issues need to be resolved before publishing:
Thank you for your very nice and critical comments, that were all very appreciated. Below, please, find our attempt to answer your main questions.
- Did the author check the bioactivity of identified peptides?
Answer: We have not yet tested the bioactivity of the peptides, but this will be performed in the future, starting with the peptide classes that were discussed in the article. In addition, we believe that the study brings new information about an understudied species in the peptidome field. From the quantification data included, we can see that even in basal conditions, there are differences in the metabolism of peptides from the same precursor protein, which may indicate a distinct proteolytic processing.
- There is no information provided on the quantification of obtained peptides.
Answer: In this new version of the manuscript after the suggestion of one of the reviewers, the data of detection of the peptides of each sample, as well as the relative quantification between the samples were performed, presented in Supplementary table 1, summarized in Figure 3 and also discussed in the main text.
- At page 2 the author should revise the nomenclature of showing ions……Several proteins related with ion transporters are expressed in the membrane cells of the gills (NKA- Na+-K+ - ATPase; NKCCl- Na+-K+-2Cl- cotransporter; NHE-Na+-H+ exchanger; H-A-H+ATPase……
Answer: One of the reviewers requested that the introduction be shortened, so the mentioned sentence was removed from the article.
- The quality of figures is poor.
Answer: Thanks. The quality of the figures have been improved in this new version of the manuscript.
- In figure 3B, the author should consider labelling the peptide fragments (y3, y4 ...) on top of the sequence corresponding to the peaks observed in the mass spectra for more clarity.
Answer: The figure indicated by the reviewer was modified. Because the space between ions is small, most b and y ions have now been indicated through tables above the MS/MS spectrum.
Round 2
Reviewer 3 Report
The authors have addressed majority of my comments. I am satisfied with this version.